# UniCTokens: Boosting Personalized Understanding and Generation via Unified Concept Tokens

**Ruichuan An**[1 2*] **Sihan Yang**[2*] **Renrui Zhang**[3†] **Zijun Shen**[4] **Ming Lu**[5] **Gaole Dai**[1] **Hao Liang**[1]
**Ziyu Guo**[3] **Shilin Yan**[1] **Yulin Luo**[1] **Bocheng Zou**[6] **Chaoqun Yang**[7] **Wentao Zhang**[1‡]

[1] Peking University   [2] Xi'an JiaoTong University   [3] CUHK   [4] Intel Labs, China
[5] Nanjing University   [6] University of Wisconsin-Madison   [7] Tsinghua University

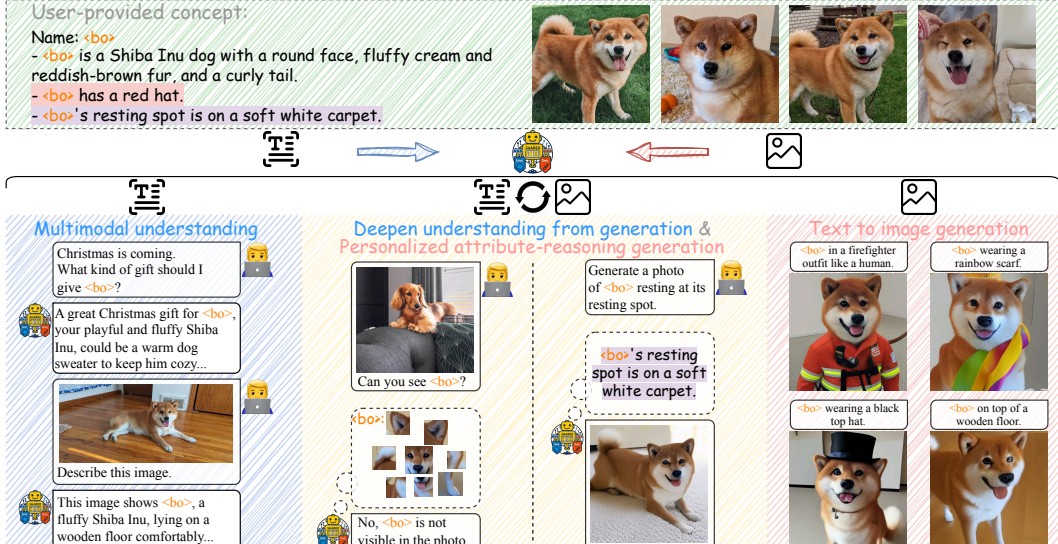

Figure 1: **The capability overview of UniCTokens.** UniCTokens achieves personalized understanding and generation of a unified VLM using user-provided concept images and texts. This is accomplished by fine-tuning a set of unified concept tokens, which harness the mutual benefits of understanding and generation. Notably, UniCTokens supports complex personalized attribute-reasoning generation, which has never been achieved by previous methods.

## Abstract

Personalized models have demonstrated remarkable success in understanding and generating concepts provided by users. However, existing methods use separate concept tokens for understanding and generation, treating these tasks in isolation. This may result in limitations for generating images with complex prompts. For example, given the concept $\langle bo \rangle$, generating "$\langle bo \rangle$ wearing its hat" without additional textual descriptions of its hat. We call this kind of generation ***personalized attribute-reasoning generation***. To address the limitation, we present UniCTokens, a novel framework that effectively integrates personalized information into a unified vision language model (VLM) for understanding and generation. UniCTokens trains a set of unified concept tokens to leverage complementary semantics, boosting two personalized tasks. Moreover, we propose a progressive training strategy with three stages: understanding warm-up, bootstrapping generation from under-

---

*Equal contribution. Email: `arctanxarc@gmail.com`

†Project Leader.

‡Corresponding author. Email: `wentao.zhang@pku.edu.cn`

standing, and deepening understanding from generation to enhance mutual benefits between both tasks. To quantitatively evaluate the unified VLM personalization, we present UnifyBench, the first benchmark for assessing concept understanding, concept generation, and attribute-reasoning generation. Experimental results on UnifyBench indicate that UniCTokens shows competitive performance compared to leading methods in concept understanding, concept generation, and achieving state-of-the-art results in personalized attribute-reasoning generation. Our research demonstrates that enhanced understanding improves generation, and the generation process can yield valuable insights into understanding. Our code and dataset will be released at: https://github.com/arctanxarc/UniCTokens.

# 1 Introduction

Personalized tasks focus on understanding and generating information that users provide. Recently, there has been growing interest in personalizing understanding models, especially in transforming general-purpose models into daily life assistants [1, 2, 3]. As a result, significant effort has been devoted to improving the personalization capabilities of Large Language Models (LLMs) [4, 5, 6] and Vision-Language Models (VLMs) [7, 8, 9, 5, 10, 11, 12]. In terms of generation, with the rapid advancement of text-to-image (T2I) models [13, 14, 15, 16, 17, 18], personalization techniques can generate highly realistic and diverse images based on user-specified concepts. They mostly employ paradigms such as test-time fine-tuning [19, 20] and retrieval augmentation [21]. LLMs, VLMs, and T2I models have shown remarkable personalization performance in their respective task domains.

Despite significant advancements in personalized generation and understanding, current methods often treat these as independent tasks, failing to effectively leverage complementary semantics [22]. Personalized understanding utilizes vision-language models (VLMs) [23, 24, 25], while personalized generation primarily employ diffusion models [26, 19, 27]. This leads to a lack of a unified personalized model for users to perform both understanding and generation. Meanwhile, personalized tasks are complex and conceptually diverse in reality, as shown in Fig. 1. For instance, when training data includes only the concept $\langle bo \rangle$, diffusion models would struggle to generate suitable images of "$\langle bo \rangle$ wearing its hat", if additional textual descriptions of its hat are provided. Additionally, concepts often include subtle visual features that are critical for identifying them. Traditional understanding models typically prioritize high-level semantic information over these subtle yet critical features [28, 29]. Ignoring the mutual semantics of the two tasks makes current methods insufficient for fully understanding and efficiently generating concepts provided by users.

Although a recent work Yo'Chameleon [22] achieves understanding and generation upon a unified VLM [30], it still presents several challenges in personalization:

- First, Yo'Chameleon utilizes distinct concept tokens for understanding and generation, whereas isolating these tasks may not fully leverage their complementary benefits.
- Second, Yo'Chameleon assesses personalized understanding and generation separately, without quantifying how understanding facilitates generation, referred as attribute-reasoning generation.

To this end, instead of fine-tuning distinct concept tokens like Yo'Chameleon, we propose UniCTokens, a personalization framework that efficiently personalizes a unified VLM by fine-tuning unified concept tokens. Additionally, we utilize a progressive training strategy with three stages, mimicking the general process of human understanding of concepts. Given a new concept, we first establish a preliminary understanding of it, and then enable the ability to visualize it through drawing, thereby enhancing the comprehension of the concept. Specifically, our method begins by warming up unified concept tokens through an understanding task. We then share the concept representations learned from this task in generative learning. Finally, during the generation process, we utilize intermediate results to provide detailed information for the understanding task. This progressive training strategy explicitly promotes mutual enhancement between personalized understanding and generation.

To better assess the personalization capabilities of the unified model, we introduce a new benchmark, UnifyBench, aimed at evaluating models' abilities in concept understanding, concept generation, and attribute-reasoning generation. When we assess our UniCTokens using UnifyBench, we consistently achieve competitive or better results than all other personalization methods. Our analysis indicates that a better understanding enhances generation, while the generation process can also provide

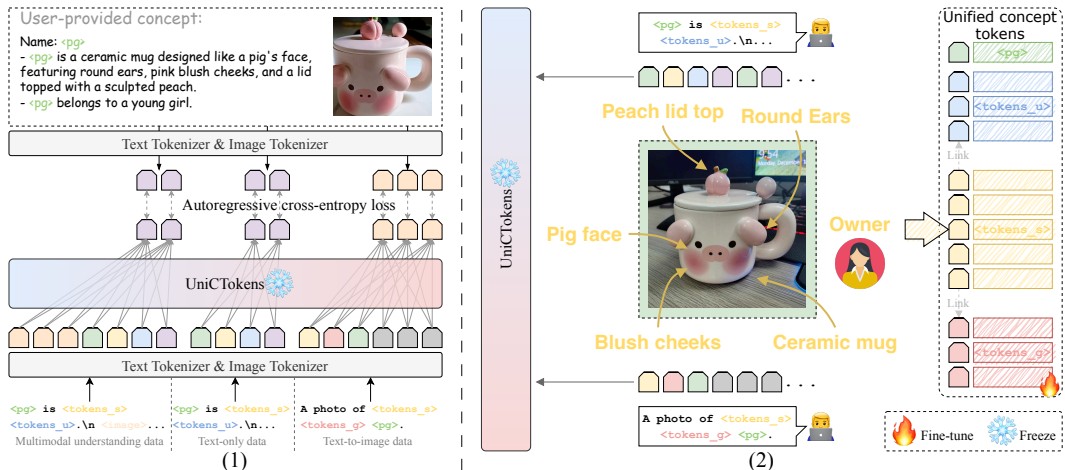

Figure 2: **The overview of UniCTokens.** Rather than training separate concept tokens for understanding and generation, we train unified concept tokens that take advantage of the mutual benefits of both tasks. Linked with shared tokens, we achieve cross-task transfer.

information that supports understanding, providing valuable insights for the development of general unified models. We believe that UnifyBench will serve as a strong foundation for future research in unified model personalization. To sum up, our contributions can be concluded as:

- We propose UniCTokens, an effective framework for personalizing unified VLMs by fine-tuning unified concept tokens instead of separate tokens for understanding and generation.

- We propose a progressive training strategy consisting of three stages to facilitate the transfer of information between tasks, promoting both personalized understanding and generation.

- We construct UnifyBench, a benchmark to evaluate concept understanding, concept generation, and attribute-reasoning generation of a personalized unified model.

- We conduct extensive experiments on UnifyBench. UniCTokens demonstrates competitive performance compared to leading methods in concept understanding and generation, achieving state-of-the-art results in attribute-reasoning generation.

## 2   Related Work

**Personalized Understanding and Generation Model.**   Model personalization mainly involves integrating concept-related information into the outputs of the model. Recent text-to-image methods depend on recontextualizing text conditions. Text inversion [26] utilizes soft prompt tuning for special token adjustments, while Dreambooth [19] modifies the entire network weights to ensure subject fidelity. Additionally, [27, 31, 32, 33, 34] inject concept-related information through varied training strategies. Large Language Models and Vision-Language Models have also witnessed trends in personalization. [35] employs a patch-based LoRA for character, while [25] utilizes a dual-tower architecture for user-aware outputs. Furthermore, [23, 24, 21] achieve personalized responses in multimodal scenarios through fine-tuning or retrieval-augmented generation, linking user information with content in images. Yo'Chameleon [22] is the first attempt at unified personalized models. However, its separate training strategy limits cross-task information sharing. We train unified concept tokens to enhance information transfer between understanding and generation tasks.

## 3   UniCTokens

We propose UniCTokens, a novel framework that efficiently manages personalized understanding and the generation of a unified VLM, as shown in Fig. 2(1). In order to transfer information between understanding and generation, we propose a three-stage training strategy: (1) Personalized understanding warm-up (2) Bootstrap generative learning from understanding and (3) Deepen understanding of

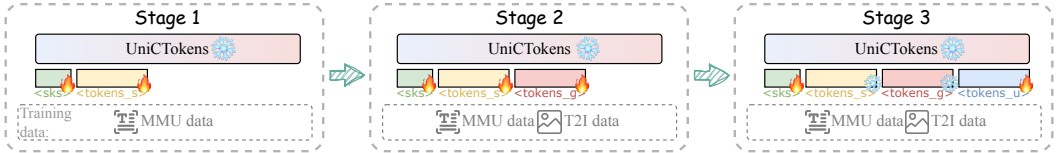

Figure 3: **Overview of Training Stages of UniCTokens.**

representation from generation. This section first defines the personalization of a unified VLM using unified concept tokens and details the three-stage training procedures.

## 3.1 Unified VLM Personalization with Unified Concept Tokens

To model the complexity of real-world personalization, we train unified concept tokens rather than using separate tokens for understanding and generation as Yo'Chameleon [22]. Given a target concept $C$, users provide personalized inputs comprising: the name of $C$, a set of images $\{I^i\}_{i=1}^n$ (typically 3 to 10) that exclusively depict $C$, and a set of textual descriptions $\{T^i\}_{i=1}^m$ containing additional attributes of $C$ that are not visually inferable (e.g., "$C$'s favorite hat is red").

The goal is to train unified concept tokens $M$ that can: (1) concept understanding: generate personalized textual responses $P_{\text{text}}$ related to $C$ (e.g., answering "What is $C$ doing in the photo?"), (2) concept generation: synthesize conditional images $P_{\text{img}}$ of $C$ under various prompts, and (3) attribute-reasoning generation: produce personalized attribute-reasoning images $P_{\text{pimg}}$ that integrate the textual information. As shown in Figure 1, the commonality between Object 2 and Object 3 lies in their classification as image generation tasks that necessitate the production of high-quality personalized concepts. The distinction between them is that the prompt for Object 2 contains only the personalized concept, whereas the prompt for Object 3 additionally incorporates certain information that is only within the understanding data (e.g., "[object Object] has a red hat" in understanding data, "a photo of [object Object] wearing its hat" for attribute-reasoning generation). Failure to leverage this information would preclude the generation of the hat in an appropriate color. Our objective is to utilize this task to measure the extent of information transfer across tasks. Detailed task and metric settings can be found in the Appendix. This process can be expressed in a formula as follows:

$$P_{\text{text}},\ P_{\text{img}},\ P_{\text{pimg}} = M\left(\{I^i\}_{i=1}^n,\ \{T^i\}_{i=1}^m\right) \tag{1}$$

## 3.2 Stage-1: Personalized Understanding Warm-up

Soft prompt tuning has proven effective in integrating new concepts for both personalized understanding and generation tasks [36, 26]. Moreover, learnable prompts are often utilized as conduits for information transfer across task [37, 38], model [39, 40], and modality [41, 42, 43]. Based on this paradigm, we represent the concept $C$ as a prompt containing learnable tokens for unified models:

$$\langle\text{sks}\rangle \text{ is } \langle\text{tokens\_s}\rangle. \tag{2}$$

where $\langle\text{sks}\rangle$ is a learnable unique identifier for this new concept and $\langle\text{tokens\_s}\rangle$ is shared tokens $\langle\text{token}_{s_1}\rangle\ldots\langle\text{token}_{s_K}\rangle$ encode semantic attributes specific to that concept. This personalized prompt serves as the system prompt during training. After understanding task training, $\langle\text{tokens\_s}\rangle$ encapsulates various characteristics of the concept (e.g., height, hairstyle, shape; see Fig. 2(2)).

To enable effective cross-task information transfer in subsequent stages, warm-up training is essential. To stabilize the training process, we adopt the token initialization strategy proposed in MC-LLaVA [24]. During training, instruction tuning is employed to optimize the initial tokens. The training dataset contains three types of Visual Question Answering (VQA) samples: (1) positive recognition VQA pairs, (2) random recognition VQA pairs, and (3) conversational VQA pairs. Detailed descriptions of the data construction process can be found in MC-LLaVA [24] Notably, user-provided textual information $\{T^i\}_{i=1}^m$ is transformed by LLMs into a set of text-only QA pairs, which are also incorporated into training. Implementation details are provided in the Appendix.

## 3.3 Stage-2: Bootstrap Generative Learning from Understanding

Training solely on understanding tasks does not equip the unified model to directly generate images containing $C$. Existing personalized unified models [22] require a substantial number of samples

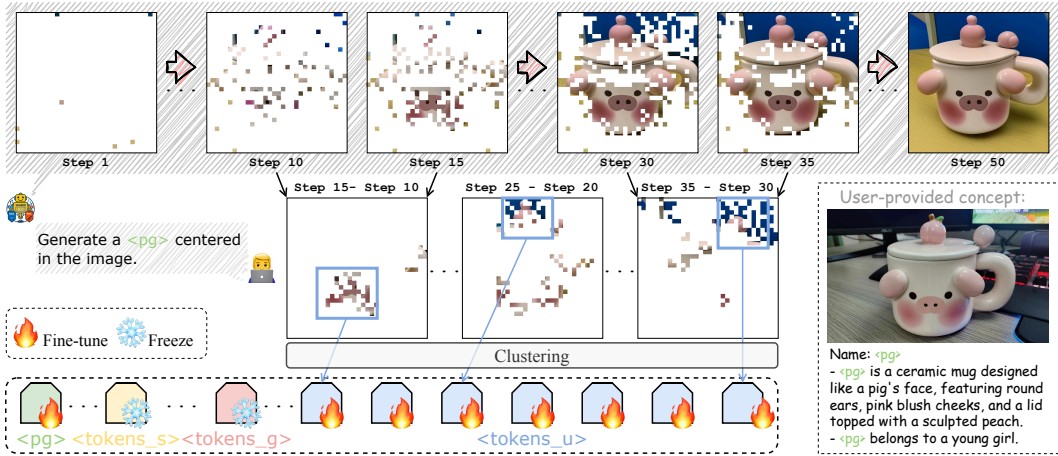

Figure 4: **Generation as Perception.** The first row depicts the generation process, while the differences at different timestamps capture concept details (e.g., pig noses and cup handles).

($\sim$1,000) to train a single concept, which is impractical for real-world applications. For humans, a preliminary understanding of concepts facilitates artistic creation. The more thoroughly humans comprehend a concept, the more accurately they can replicate it in their artwork. Thus, we aim to explore the potential of leveraging understanding information to facilitate generation.

Directly training generation tasks on shared tokens presents a straightforward strategy. However, this direct training approach results in the model losing its general conditioning capability on $C$ and significantly diminishes performance on understanding tasks. We posit that this discrepancy is due to variations in task distributions, which encompass specific information that cannot be directly shared. Thus, we integrate new tokens to enhance model training tailored for text-to-image (T2I). Inspired by DreamBooth [19], the prompt for Stage 2 of T2I training is formalized as follows:

$$\text{A photo of } \langle \text{tokens\_s} \rangle \langle \text{tokens\_g} \rangle \langle \text{sks} \rangle. \tag{3}$$

where $\langle \text{tokens\_g} \rangle$ is $\langle \text{token}_{g_1} \rangle ... \langle \text{token}_{g_M} \rangle$, consisting $M$ learnable tokens. The token initialization strategy is outlined as follows: If the concept is human-related, the features acquired from understanding tokens inherently capture aspects of style, preferences, and overall appearance. To better maintain the concept characteristic, inspired by [44, 45], a facial encoding model is employed to derive embeddings for initializing the tokens. If the concept is non-human, we simply apply the same method mentioned in Stage 1, due to the lower complexity of generating non-human concepts.

Leveraging priors from understanding within shared tokens, UniCTokens can generate images that effectively retain concept features with only $3 \sim 10$ training samples and transfer extra information into generation, thereby demonstrating its efficiency and benefiting from understanding.

### 3.4 Stage-3: Deepen Understanding Representation from Generation

Existing literature [28] shows that understanding models focus primarily on high-level semantic information, while the generation model is more responsive to low-level features. Experimental findings indicate that the training in Stage 1 may not effectively address tasks related to low-level characteristics, such as fine-grained recognition. After completing the generation training, model is capable of producing visually similar images, which may have the potential to enhance its understanding capabilities through effective utilization of information derived from generation process.

Show-o [46] utilizes a decode methodology following MaskGIT [47], removing masked tokens to derive the final image. To generate an image $x_T$, the process can be formularized as follows:

$$p_\theta(x_0|x_T) = \prod_{t=1}^{T} p_\gamma(x_{t-1}|x_t) \tag{4}$$

where $x_t$ denotes the latent variables at step $t$ and $p_\theta(x_{t-1}|x_t)$ represents the conditional probability distribution defined by the model parameters $\gamma$. Visualization of the process is shown in the top row of Fig. 4. We also present the results of image differencing between different time steps.

Experimental observations reveal that Show-o typically generates images of the target concept $C$ in a coarse-to-fine manner, starting with subject components and gradually completing the background. Semantically meaningful subject features (e.g., human eyes) begin to emerge as early as the first $\frac{1}{5}$ of the timesteps, while background generation generally starts around $\frac{2}{3}$ of the diffusion process.

The subject is generated gradually in distinct components. This process reflects the model's assessment of the relative difficulty of different parts of the concept. Areas where the model has higher confidence are prioritized for earlier generation, while regions with greater uncertainty are produced later. We analyze the differences between intermediate results at various timestamps to identify areas where the model encounters more challenges. To improve the localization of regions, we utilize the k-means algorithm to determine the most valuable visual tokens:

$$v_r = set(\text{k-means}(x_m - x_n)), \ m \in \tilde{m}, n \in \tilde{n} \tag{5}$$

where $v_r$ represents relative hard regions selected by the model and $\tilde{m}$ and $\tilde{n}$ represent discrete time steps, where $\tilde{m}$ denotes a later timestamp than $\tilde{n}$. These regions contain fine-grained concept information and will serve as initialization for the newly added tokens, denoted as $\langle \text{tokens\_u} \rangle = \langle \text{token}_{u_1} \rangle \ldots \langle \text{token}_{u_N} \rangle$. The final prompt for concept $C$ in understanding tasks is as follows:

$$\langle \text{sks} \rangle \text{ is } \langle \text{tokens\_s} \rangle \ \langle \text{tokens\_u} \rangle. \tag{6}$$

During Stage 3, understanding samples are exclusively constructed from recognition VQA pairs to further strengthen the model's identification capability. In parallel, the T2I training data continues to update the concept identifier $\langle \text{sks} \rangle$. At this stage, the only learnable tokens are $\langle \text{sks} \rangle$ and $\langle \text{tokens\_u} \rangle$.

This strategy enhances the model's ability to identify and extract critical details requiring improvement during the generation process. The understanding task lacks perception of low-level concept details, which leads to complementary information provided by the generation.

## 4 Experiment

### 4.1 Experiment Setup

**Implement Details.** We set the number of learnable tokens as $K = 16$, $M = 8$, and $N = 8$ respectively. All training stages are optimized using AdamW and each stage is trained for 20 epochs. The batch size is set to 4 for understanding tasks in stage 1, and 1 for both stage 2 and stage 3, as well as for T2I generation. All experiments are conducted on A800 GPUs. For the backbone, we adopt Show-o512x512 [46] as the base model. More training details can be found in the Appendix.

**Dataset.** We collect Unifybench of 20 concepts: Person (10), Pets (5), Objects (5). Each concept is associated with 10~15 images for training and testing. In addition, each concept is annotated with 1~2 extra attributes that are not visually inferable from training images (e.g., "this person owns a dog"). To comprehensively evaluate both understanding and T2I capabilities, We design specific test data for each task. This includes standard understanding QA pairs and generation prompts, as well as personalized attribute-reasoning queries. Please refer to Appendix for more details on our dataset.

**Baselines.** Our approach is primarily evaluated against four distinct categories of methods. The most direct comparison utilizes a unified base model incorporating personalized text and image prompts. Show-o's inability to support interleaved image prompts precluded this comparison. The textual prompts are derived from the captions generated by GPT-4o for each concept and subsequently summarized to meet the required token length. Another significant baseline for comparison is the recent unified customized model, Yo'chameleon [22]. We retrain the model according to the original paper, utilizing 1,000 images per concept and a 7B base model. Additionally, we evaluate the current GPT-4o on our benchmark to serve as an upper bound. Lastly, we conduct comparisons between models focused solely on understanding or generation. More details are provided in the Appendix.

**Metrics.** For personalized attribute-reasoning image generation, we use VLMs to score the generated images (from 0 to 1) based on their consistency with the extra attributes of concepts embedded in the prompt. Metrics for separate personalized understanding and generation are detailed in Appendix. Final results are obtained by averaging the scores across all concepts for each evaluation metric.

| Type | Methods | Model Size | Token | Training Images | Personalized Understanding | | | | | Personalized Generation | | | | PARG | |
|------|---------|------------|-------|-----------------|------|------|-----|------|-----|------|--------|------|-----------|-------|--------|
| | | | | | Rec. | VQA | | QA | | Pure Gen. | | | People Gen. | | |
| | | | | | Weight | BLEU | GPT | BLEU | GPT | CLIP-I | CLIP-T | DINO | Face-Simi | Score | CLIP-I |
| **Upper Bound** | GPT-4o+TP | 200B | ~100 | - | 0.742 | 0.473 | 0.676 | 0.610 | 0.685 | 0.689 | 0.301 | 0.626 | 0.198 | 0.780 | 0.690 |
| | GPT-4o+IP | 200B | ~1,000 | - | 0.773 | 0.543 | 0.685 | 0.589 | 0.652 | 0.794 | 0.310 | 0.722 | 0.559 | 0.782 | 0.792 |
| | Real Images | - | - | - | - | - | - | - | - | 0.832 | - | 0.728 | 0.739 | - | 0.832 |
| **Und. Only** | Yo'LLaVA | 13B | 16 | ~100 | 0.919 | 0.609 | 0.629 | 0.612 | 0.593 | - | - | - | - | - | - |
| | MC-LLaVA | 13B | 16 | ~10 | 0.924 | 0.628 | 0.637 | 0.601 | 0.583 | - | - | - | - | - | - |
| | RAP-MLLM | 13B | ~1,000 | - | 0.940 | 0.616 | 0.616 | 0.712 | 0.722 | - | - | - | - | - | - |
| | Qwen2.5-VL + TP | 3B | ~100 | - | 0.660 | 0.407 | 0.727 | 0.574 | 0.774 | - | - | - | - | - | - |
| | Yo'LLaVA(Phi-1.5) | 1.3B | 16 | ~100 | 0.765 | 0.488 | 0.497 | 0.510 | 0.494 | - | - | - | - | - | - |
| **Gen. Only** | Text inversion | 1.0B | - | ~10 | - | - | - | - | - | 0.630 | 0.247 | 0.569 | 0.371 | 0.070 | 0.628 |
| | DreamBooth (SD) | 1.0B | - | ~10 | - | - | - | - | - | 0.649 | 0.281 | 0.591 | 0.436 | 0.071 | 0.650 |
| **Unified Model** | Chameleon+TP | 7B | ~100 | - | 0.690 | 0.413 | 0.488 | 0.509 | 0.564 | 0.547 | 0.176 | 0.509 | 0.011 | 0.329 | 0.549 |
| | Chameleon+IP | 7B | ~1,000 | - | 0.493 | 0.445 | 0.498 | 0.407 | 0.535 | 0.523 | 0.160 | 0.469 | 0.066 | 0.299 | 0.499 |
| | Show-o+TP | 1.3B | ~100 | - | 0.566 | 0.461 | 0.409 | 0.504 | 0.579 | 0.664 | 0.264 | 0.553 | 0.048 | 0.770 | 0.660 |
| | Yo'Chameleon | 7B | 32 | ~1,000 | 0.764 | 0.474 | 0.507 | 0.510 | 0.581 | 0.697 | 0.236 | 0.590 | 0.224 | 0.266 | 0.698 |
| | Ours | 1.3B | 32 | ~10 | 0.790 | 0.503 | 0.523 | 0.544 | 0.603 | 0.750 | 0.282 | 0.646 | 0.334 | 0.359 | 0.749 |

Table 1: **Quantitative Comparison on UnifyBench.** TP = Text Prompt. IP = Image Prompt. PARG = Personalized Attribute-Reasoning Generation. The best and second best are highlighted.

## 4.2 Our Unified Personalized Benchmark (UnifyBench)

**Personalized Concept Understanding.** This task requires responding to queries with concept identifiers and images. Following [23, 36, 24], we conduct experiments on personalized recognition, VQA, and QA tasks. As demonstrated in Tab. 1, our proposed UniCTokens significantly enhances the performance of the vanilla Show-o model by an average of 8.9%, while utilizing fewer tokens. Notably, when compared with unified models with a much larger number of parameters, our model also achieves decent performance over all understanding tasks while keeping a smaller training sample(~10 v.s. ~1,000). Given such promising results, we envision UniCTokens as a potential next-generation paradigm for unified personalized understanding.

**Personalized Concept Generation.** Personalized image generation is more challenging than language generation. It requires controlling many pixels with a few tokens that contain conceptual information. As shown in Tab. 1, we achieve state-of-the-art results in personalized generation across three evaluation metrics among unified models. Our method also outperforms the unified models in individual generation, producing realistic faces with conceptual features. Utilizing image prompts, GPT-4 demonstrates effectiveness, but it requires a large number of tokens. Simply adding image tokens does not guarantee improvements, as the effectiveness also depends on the model's performance, as demonstrated in Chameleon. Fig. 5 showcases the generated image of UniCTokens, illustrating the consistency of conditional control generation and concept-related features.

**Personalized Attribute-Reasoning Generation.** This task evaluates the capability of unified models to generate images with additional textual personalized knowledge (e.g., "⟨ sks ⟩ likes playing with a yellow ball."). This task is challenging because the training data for T2I does not include this information. Thus, the model must possess a capacity for cross-task information transfer to handle this challenge. As shown in Tab. 1, while directly utilizing text prompts appears to improve the T2I scores, it decreases in CLIP-I, indicating a quality reduction of generated images. Our model achieves optimal results in balancing these two aspects, effectively incorporating additional personalized knowledge into generated images while preserving the characteristics of the concept. Qualitative results in Fig. 5 show that generated picture can precisely reflect the textual description.

## 4.3 Existing Personalized Understanding and Generation Benchmarks

In order to facilitate a fair comparison, we also evaluated our method on benchmarks for pure personalized understanding and generation. The results of the evaluation on Yo'LLaVA [36] and MC-LLaVA Datasets [24] are presented in Tab. 2 (left). Due to the limitations of the unified model's capabilities on multi-concept tasks, we only conducted tests on the single-concept portion of MC-

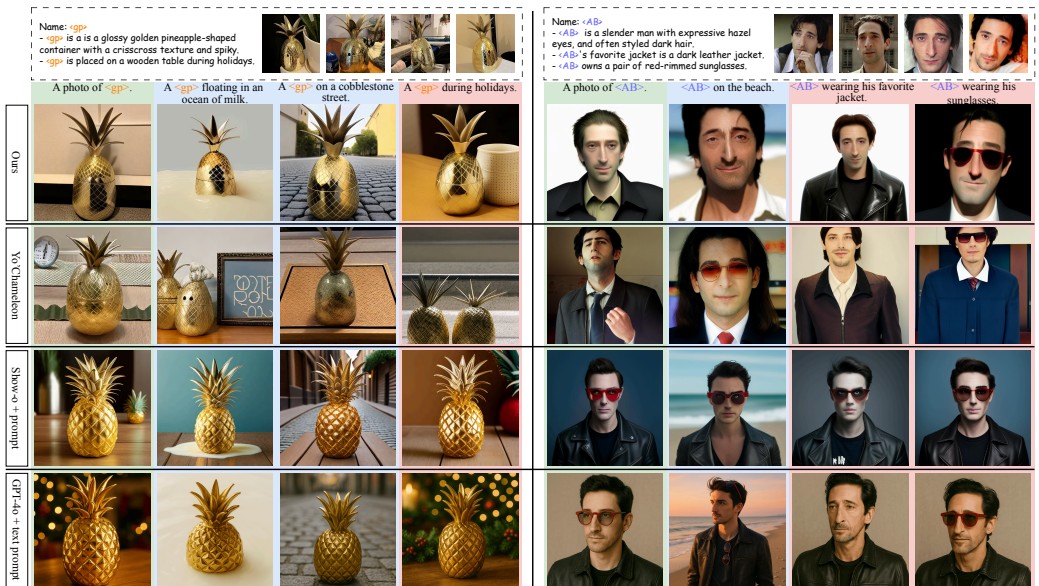

Figure 5: **Qualitative Comparisons among UniCTokens, Yo'Chameleon and GPT-4o.** Our proposed UniCTokens demonstrates its controllable and personalized generation.

| Type | Method | Model Size | Token | Training Images | Yo'LLaVA Rec Weight | Yo'LLaVA VQA Acc | Yo'LLaVA QA Acc | MC-LLaVA Rec Weight | MC-LLaVA VQA BLEU | MC-LLaVA QA Acc |
|------|--------|------------|-------|-----------------|------|------|------|------|------|------|
| Und. Only | LLaVA+TP | 13B | ~50 | - | 0.819 | 0.913 | 0.803 | 0.594 | 0.428 | 0.597 |
| | Yo'LLaVA | 13B | 16 | ~100 | 0.924 | 0.929 | 0.883 | 0.841 | 0.643 | 0.703 |
| | MC-LLaVA | 13B | 16 | ~10 | 0.947 | 0.941 | 0.910 | 0.912 | 0.679 | 0.723 |
| | Qwen2.5-VL+TP | 3B | 32 | - | 0.671 | 0.873 | 0.709 | 0.621 | 0.423 | 0.562 |
| | Yo'LLaVA(Phi-1.5) | 1.3B | 16 | ~100 | 0.792 | 0.613 | 0.726 | 0.714 | 0.512 | 0.603 |
| Unified Model | Chameleon+TP | 7B | ~64 | - | 0.727 | 0.523 | 0.716 | 0.637 | 0.421 | 0.662 |
| | Yo'Chameleon | 7B | 32 | ~1,000 | 0.845 | 0.604 | 0.721 | 0.741 | 0.597 | 0.670 |
| | Show-o+TP | 1.3B | ~64 | - | 0.691 | 0.513 | 0.591 | 0.601 | 0.587 | 0.469 |
| | Ours | 1.3B | 32 | ~10 | 0.852 | 0.615 | 0.738 | 0.754 | 0.630 | 0.679 |

| Type | Method | Model Size | Token | Training Images | DreamBench CLIP - I | DreamBench CLIP - T | Yo'LLaVA CLIP - I |
|------|--------|------------|-------|-----------------|--------|--------|--------|
| Gen. Only | Real Images | - | - | - | 0.885 | - | 0.851 |
| | DreamBooth | 1.0B | - | ~10 | 0.701 | 0.283 | 0.632 |
| | DreamBooth | 1.0B | - | ~3000 | 0.803 | 0.305 | 0.800 |
| | Text inversion | 1.0B | - | ~10 | 0.687 | 0.271 | 0.619 |
| Unified Model | Chameleon+TP | 7B | ~100 | - | 0.599 | 0.180 | 0.566 |
| | Chameleon+IP | 7B | ~1,000 | - | 0.581 | 0.159 | 0.487 |
| | Show-o+TP | 1.3B | ~100 | - | 0.690 | 0.247 | 0.665 |
| | Yo'Chameleon | 7B | 32 | ~1,000 | 0.795 | 0.225 | 0.783 |
| | Ours | 1.3B | 32 | ~10 | 0.800 | 0.287 | 0.794 |

Table 2: **Performance on Personalized Understanding and Generation Benchmarks.** TP = Text Prompt. IP = Image Prompt. The best and second best performances are highlighted.

LLaVA. Our method outperforms the current leading unified personalized model, utilizing only 1.3 B parameters and fewer training images. Notably, UniCTokens outperforms on all understanding tasks with an average of 5.13% when compared to an important baseline, Yo'LLaVA(1.3B), demonstrating the potential of scaling UniCToken to achieve state-of-the-art performance.

We evaluate personalized generation capabilities of UniCTokens on Dreambench [19] and Yo'LLaVA Dataset [36]. Our approach significantly enhances the capabilities of vanilla Show-o. Additionally, our performance surpasses that of Yo'Chameleon, particularly on CLIP-T, which measures the model's ability for controllable generation. This improvement can be attributed to our more effective prompt design and mutual enhancement between personalized tasks. Notably, we also outperform the significant generative baseline, Dreambooth, with the same training data, demonstrating that our method can generate high-quality and realistic images containing user-provided concepts.

## 4.4 Ablation Study and Analysis

**Better Understanding Leads to Better Generation.** Through modulation of training epochs and data requirements, we developed unified models with varying understanding capacities. Models' understanding capabilities benefit from increased training epochs and data. Fig. 6 (left) shows that as the understanding capability of the unified models improves, their generative ability also enhances. Moreover, our analysis shows that factors influencing enhanced understanding have different effects on generative performance, as indicated by the varying slopes. For models with insufficient training

data, limitations on generative capabilities may not primarily result from the duration of training. This finding could offer crucial insights for future research on general unified models.

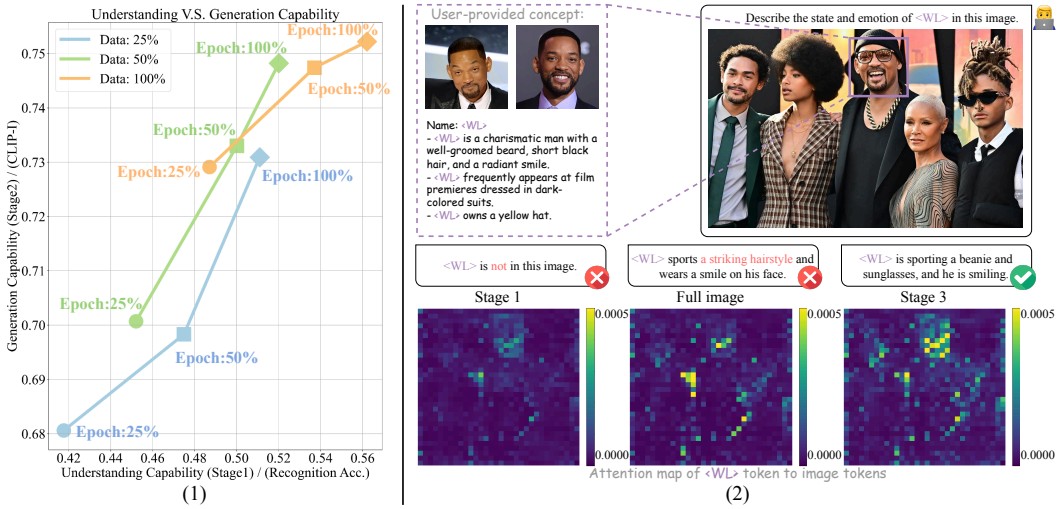

Figure 6: (1) Ablation study on relationship between personalized understanding and generation. (2) Visualization of different token initialization methods for stage 3 and their corresponding model outputs. With generation process as perception, UniCTokens focuses more on the concept.

**Generation Process as Perception for Understanding.** The process of identifying challenging regions through generation can be conceptualized as a form of perception. To investigate how this process facilitates understanding, we evaluated four main baselines: (1) Pure stage 1, without additional generative assistance; (2) Direct finetuning with the same data; (3) Utilizing the entire image for token initialization; (4) Initialization via randomly selected patches. As illustrated in Tab. 3 (right), additional training only brings margin improvements, while general initialization methods still fall short of the performance achieved by our proposed generation as a perception process. Fig. 6 (right) indicates that models trained with our method are more likely to produce detailed sentences. The attention map of UniCTokens demonstrates a greater focus on the provided concepts, reflected by higher scores, whereas utilizing the full image for initialization introduces the issue of attention dispersion, resulting in uneven distribution. These suggest that generative priors may be effectively integrated into the understanding component, resulting a better understanding for concepts.

| Stage | Personalized Understanding | | | | | Personalized Generation | | | | | PARG | |
| | Rec. | VQA | | QA | | Pure Gen. | | | People Gen. | | | |
| | Weight | BLEU | Score | BLEU | Score | CLIP - I | CLIP-T | DINO | Face-Simi | | Score | CLIP-I |
|---|---|---|---|---|---|---|---|---|---|---|---|---|
| Stage 1 | 0.637 | 0.494 | 0.538 | 0.540 | 0.600 | 0.524 | 0.278 | 0.446 | 0.060 | | 0.204 | 0.511 |
| Stage 2 w/o 1 | 0.616 | 0.479 | 0.488 | 0.527 | 0.581 | 0.695 | 0.243 | 0.582 | 0.210 | | 0.297 | 0.695 |
| Stage 2 | 0.621 | 0.497 | 0.532 | 0.538 | 0.605 | 0.752 | 0.280 | 0.648 | 0.349 | | 0.349 | 0.750 |
| Stage 3 | 0.790 | 0.503 | 0.523 | 0.544 | 0.603 | 0.750 | 0.282 | 0.646 | 0.334 | | 0.334 | 0.749 |

| Init strategy | Rec. | VQA | QA |
| | Weight | Score | Score |
|---|---|---|---|
| Stage 1 | 0.637 | 0.538 | 0.427 |
| No Init | 0.670 | 0.520 | 0.429 |
| Full Image | 0.723 | 0.529 | 0.423 |
| Random Patch | 0.681 | 0.520 | 0.421 |
| Stage 3 | 0.790 | 0.523 | 0.431 |

Table 3: Performance of different training stages (left). Different initialization strategies (right).

**Different Training Strategies.** Our approach utilizes a three-stage training strategy to facilitate information transfer across tasks. We then validate this design, beginning from generation, especially examining the inclusion of Stage 1. Omitting Stage 1 yields a variant of Text inversion distinguished by an increasing parameter count. Tab. 3 emphasizes the essential role of Stage 1, particularly in personalized attribute-reasoning generation, omitting Stage 1 leads to significant degradation in generation quality. This further corroborates the existence of information transfer across tasks in our design. Stage 3 does not update the generative parameters, resulting in a subtle influence on the generation task. In the context of the understanding task, we present the results of the evaluations conducted across all stages. Although performance in the understanding task declines in Stage 2 to facilitate generation, the subsequent injection of fine-grained visual information from Stage 3 enhances understanding capabilities. Results demonstrate that our approach achieves optimal performance, facilitating mutual enhancement between the two tasks.

# 5   Conclusion

We present UniCTokens, an innovative framework designed to personalize a unified VLM by training unified concept tokens. Our proposed three-stage training strategy enables UniCTokens to efficiently enhance personalized understanding and generation, facilitating cross-task information transfer to achieve mutual enhancement. Experimental results demonstrate that, while maintaining a smaller model size and fewer training samples, UniCTokens achieves state-of-the-art performance in concept understanding, concept generation, and attribute-reasoning generation tasks. The insights derived from our analysis are noteworthy, shedding light on the development of unified VLMs. Furthermore, the advancement of personalized AI holds significant promise for improving human lives by enabling tailored interactions, enhancing creativity, and offering solutions that align with individual needs and preferences, thereby fostering a more intuitive and responsive technological ecosystem.

# 6   Acknowledgments

This work is supported by the National Key R&D Program of China (2024YFA1014003), National Natural Science Foundation of China (92470121, 62402016), and High-performance Computing Platform of Peking University.

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

# A  Additional Implement Detail

**Loss.**  We use a standard autoregressive loss based on masked language modeling. Given a training instance $(X_q, X_a)$—where $X_q$ denotes the question and $X_a$ the answer—we apply the standard masked language modeling loss to compute the likelihood of $X_a$:

$$\mathcal{L}(X_a \mid X_q, \theta) = -\sum_{t=1}^{T} \log P(X_{a,t} \mid X_q, X_{a,<t}, \theta) \tag{7}$$

Here, $T$ is the length of the answer $X_a$, $X_{a,t}$ denotes the $t$-th token, and $P(X_{a,t} \mid X_q, X_{a,<t}, \theta)$ is the probability of predicting $X_{a,t}$ given the image $I$, question $X_q$, and preceding tokens $X_{a,<t}$.

**Stage Configuration.**  We summarize the full three-stage training setup in Tab. 4. Unless otherwise stated, before Stage 3 we run $k$-means with $K = 2$, using cosine distance ($d(\mathbf{u}, \mathbf{v}) = 1 - \frac{\mathbf{u}^\top \mathbf{v}}{\|\mathbf{u}\|\|\mathbf{v}\|}$). At each selection step, we retain the cluster containing the largest number of patches.

|  | Stage 1 | Stage 2 | Stage 3 |
|---|---|---|---|
| **Training Data** | Positive recognition VQA pairs Random recognition VQA pairs Conversational (V)QA pairs | Positive recognition VQA pairs Random recognition VQA pairs Conversational (V)QA pairs T2I data | Positive recognition VQA pairs Random recognition VQA pairs T2I data |
| **Trainable Tokens** | $\langle$sks$\rangle$; $\langle$tokens$_s\rangle$ | $\langle$sks$\rangle$; $\langle$tokens$_s\rangle$; $\langle$tokens$_g\rangle$ | $\langle$sks$\rangle$; $\langle$tokens$_u\rangle$ |
| **Batch Size - T2I** | N/A | 1 | 1 |
| **Batch Size - MMU** | 4 | 1 | 1 |
| **Learning Rate** | $1\times10^{-4}$ | $1\times10^{-3}$ | $1\times10^{-4}$ |
| **Epoch** | 20 | 20 | 20 |
| **Optimizer** | AdamW | AdamW | AdamW |

Table 4: Multi-stage training configuration.

The MMU training data used for Stage 2 is essential to maintaining the model's conversational abilities. The presence or absence of these data has minimal impact on model's generative capabilities.

# B  Additional Experiment Setup

**Metrics.**  For understanding, we evaluate the model on three tasks: personalized recognition, VQA, and QA. In the Recognition task, we compute the recall for both positive and negative samples and report their arithmetic mean. For VQA and QA, we evaluate the predicted answers using BLEU [48] scores against ground truth, and further apply an LLM-based evaluation that scores responses from 0 to 1 based on alignment with key points in the reference answers. For general personalized image generation, we adopt prompts from the DreamBooth [19] dataset, and evaluate image quality using CLIP-based metrics: CLIP-I (image-to-image similarity) and CLIP-T (image-to-text similarity). Since half of our concepts are human subjects, we additionally employ the off-the-shelf ArcFace model [49] to measure facial similarity between generated and reference images. For the evaluated GPT-4o and GPT-4o used for scoring, we utilize different versions of them to make fair comparisons.

**Comparing Baselines.**  We supplement the baselines not described in the main text:

- **LLaVA+Prompt**: We first prompt LLaVA to generate captions for all training images of a concept, then prompt LLaVA to summarize the captions into a concise, personalized description. During inference, we add relevant captions to the input to supply concept-specific information.

- **Yo'LLaVA** [36]: One of the earliest work to explore VLM personalization. Following the original paper, we manually construct hard negative datasets and train Yo'LLaVA with different sizes of base model(Phi-1.5 [50], 1.3B and Vicuna [51], 13B) to make a fair comparison.

- **MC-LLaVA** [24]: A model designed for enhancing multi-concept personalization tasks. Utilizing dual textual and visual prompts, it serves as a strong baseline for personalized understanding.

- **RAP-MLLM** [21]: We utilize the RAP-LLaVA model and follow the RAP-MLLM approach to construct a personalized database for each concept. To obtain the capability of understanding personalization, RAP-MLLM conducted a post-training on a dataset of 260K in size.

- **Dreambooth** [19]: DreamBooth enhances the personalization capability of generative models by allowing users to fine-tune models with a limited number of images, thereby producing highly specific and context-aware outputs. For fair comparison, we utilize different number of training data (10, 3,000) to better evaluate Dreambooth.

- **Text Inversion** [26]: Text inversion is a technique that transforms textual prompts into corresponding visual representations, enabling the generation of images based on description.

## C  Catastrophe Forgetting

When a model acquires new knowledge, there exists the risk of catastrophic forgetting of previously learned information. Following Yo'Chameleon [22], we evaluate the personalized model on several benchmarks assessing general capabilities. We conducted a comparative analysis against the original Show-o across well-established multimodal benchmarks: GenEval [52], MMMU [53], and POPE [54]. The experimental results shown in Fig. 7 indicate that, despite training through a three-stage process and having task-specific tokens, model performance does not diminish after each stage, effectively preserving the model's general capabilities.

|          | GenEval | MMMU | POPE |
|----------|---------|------|------|
| Original | 0.68    | 26.7 | 80.0 |
| Stage 1  | 0.67    | 26.6 | 80.0 |
| Stage 2  | 0.66    | 26.6 | 80.0 |
| Stage 3  | 0.66    | 26.6 | 80.0 |

Figure 7: **Catastrophic Forgetting Evaluation.** UniCTokens maintains performance similar to vanilla Show-o.

Figure 8: **Limitations.** Example images under different style controls.

## D  Additional Related Work

**Unified Understanding and Generation.**   A multitude of efforts have been dedicated to employing a single model for both understanding and generation. SEED [55] adapted image representation through discrete tokenization for language modeling, leveraging autoregressive conditioning for generation via an external decoder. [56, 57] restructured conditioning information aggregation but still relied on extra modules for generation. Chemeleon [30] is an early hybrid unified model capable of generating and understanding intert-leaved text-image content. Janus [28] claims that understanding and generation require distinct information, employing different tokenizers for each task. Emu3 [58] converts images, text, and video into discrete tokens, enabling joint training of a Transformer on multimodal sequences. [46, 59] utilize a hybrid of autoregressive and diffusion methods for text and image processing. The above-mentioned work all focuses on general tasks, neglecting exploration in personalization scenarios. In this paper, we employ Show-o [46] as the backbone to efficiently achieve unified personalization without forgetting the general capability.

**Bridging Understanding and Generation.**   The establishment of unified models seeks to optimize the strengths of understanding and generation, allowing information transfer between tasks and mutual improvement. Early efforts to link these tasks mainly utilized a serial processing paradigm. [60, 61, 62] employed image captioning models to generate textual conditions for text-to-image models. While these methods highlight the potential of leveraging understanding for generation, they lack end-to-end optimization and frequently suffer from information loss. [63, 64, 65, 66, 67] explored deeper integration of understanding and generation. MetaQuery [68] utilizes learnable queries on frozen VLMs to extract conditions for generation, but it primarily emphasizes how

understanding aids generation while neglecting the inverse. In this paper, we propose a three-stage training strategy that achieves information transfer and mutual enhancement between tasks in personalization scenarios, providing insights for the broader development of unified models.

**Reasoning in Understanding and Generation.** The success of Deepseek-R1 [69] promotes the exploration of reasoning. The recent work focuses on designing new reward [70, 71, 72], constructing valuable samples [73, 74, 75] and building reasonable reinforcement learning algorithm [76, 77, 78] to fully unleash the potential of LLMs [79] and VLMs [80, 81, 82]. Many works [83, 84] tend to transfer the reasoning capability into image generation process. [15] analysis the effectiveness of Chain-of-Thought(CoT) in generation, while [16] utilizing semantic- and token-level CoT to handle complex reasoning scenarios. Inspired by there work, we propose complex personalized attribute-reasoning generation, which better modeling the real world user query.

# E    Additional Qualitative Results

Figure 9: More qualitative results generated from our model.

# F    Additional Quantitative Experiments

**Learnable Token Length.** We systematically varied the quantity of learnable tokens, as illustrated in Fig.10. As the number of learnable tokens increases, the model's performance in personalized understanding and generation improves. This improvement is attributed to the increased parameter count providing more learning capacity. However, simply increasing the number of parameters is not always beneficial. When the parameter count becomes excessive (e.g., 64 tokens), the CLIP-T score declines, which may be due to excessively long contexts that make conditioning more difficult. The improvement rates vary across different tasks, reflecting a shift in task domains.

**Cost of Adding a New Concept.** Tab. 5 reports per-concept FLOPs and wall-clock time under a unified setup. Methods that personalize with large per-concept image sets—exemplified by Yo'Chameleon (about 1,100 images)—incur the highest cost ($7 \times 10^{17}$ FLOPs; 120 min), whereas

| Learnable Token Length | Und. | | Gen. | |
|---|---|---|---|---|
| | Rec. | | Pure Gen. | |
| | Weight | | CLIP-I | CLIP-T |
| 0 (only <sks>) | 0.608 | | 0.677 | 0.271 |
| 1 (1+0+0) | 0.641 | | 0.680 | 0.274 |
| 4 (2+1+1) | 0.682 | | 0.688 | 0.275 |
| 8 (4+2+2) | 0.727 | | 0.707 | 0.277 |
| 16 (8+4+4) | 0.754 | | 0.731 | 0.279 |
| 32 (16+8+8) | 0.790 | | 0.750 | 0.282 |
| 64 (32+16+16) | 0.793 | | 0.763 | 0.271 |

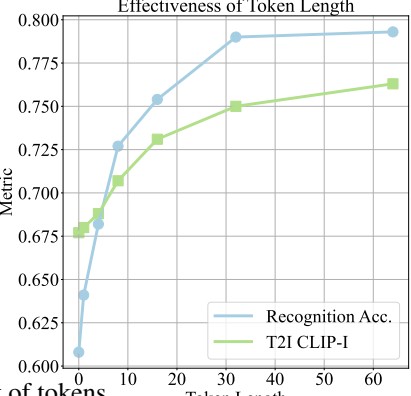

Figure 10: Ablations on amount of tokens.

generator-only tuning (DreamBooth) is far cheaper ($2 \times 10^{15}$ FLOPs; 7 min) but limited in scope. Among unified models, UniCTokens achieves a favorable cost–capability trade-off: $1.3 \times 10^{17}$ FLOPs and 25 min per concept, yielding lower time than Yo'Chameleon.

| Type | FLOPs / Concept | Time / Concept |
|---|---|---|
| Yo'LLaVA-13B | $1.5 \times 10^{17}$ | 30 min |
| Yo'LLaVA-Phi | $1.6 \times 10^{16}$ | 10 min |
| DreamBooth (SD) | $2 \times 10^{15}$ | 7 min |
| Yo'Chameleon | $7 \times 10^{17}$ | 120 min |
| UniCTokens (ours) | $1.3 \times 10^{17}$ | 25 min |

Table 5: Per-concept training cost.

**Training Scheme: Three-Stage vs. Joint.** Tab. 6 contrasts a single-stage joint baseline with our three-stage schedule. The staged schedule delivers consistent gains in personalized understanding and generation—with the largest improvements on personalized attribute-reasoning generation. These results indicate that staging is not redundant: an understanding warm-up first stabilizes concept bindings; generation bootstrapped from these bindings improves conditioning quality; the final stage feeds generation signals back to understanding. Consequently, cross-task transfer is strengthened without increasing model capacity.

| Method | Rec. Weight | VQA GPT | QA GPT | Pure Gen. CLIP-I | Pure Gen. CLIP-T | People Gen. Face-Sim. | PARG Score | PARG CLIP-I |
|---|---|---|---|---|---|---|---|---|
| Joint Training | 0.709 | 0.489 | 0.565 | 0.681 | 0.258 | 0.288 | 0.269 | 0.678 |
| 3-Stage (Ours) | 0.790 | 0.523 | 0.603 | 0.750 | 0.282 | 0.334 | 0.359 | 0.749 |

Table 6: Joint vs. three-stage. PARG = Personalized Attribute-Reasoning Generation.

**Judge Models and Human Alignment.** Beyond GPT-based scoring and the classical metrics reported in the main paper, we additionally evaluate with Gemini-2.5-Pro and a human study on 300 samples (five per concept per task). As shown in Tab. 7, the relative ordering of methods is consistent between Gemini and human judges across VQA, QA, and personalized attribute-reasoning generation: UniCTokens ranks highest where reported. These trends support using LLM judges as a practical proxy while remaining aligned with human preferences.

# G    Details of Dataset

The dataset's data sources comprise animals and objects obtained from MC-LLaVA [24], Yo'LLaVA [36] and MyVLM [23], with images of individuals sourced from Yo'LLaVA and various online platforms. All images and generated training data have been subjected to rigorous human validation processes. We present a set of images along with extra textual information in Fig. 11.

| Gemini-2.5-Pro | | | | | Human Study | | | |
|---|---|---|---|---|---|---|---|---|
| Methods | VQA | QA | PARG | | Methods | VQA | QA | PARG |
| Yo'LLaVA–Phi | 0.492 | 0.498 | — | | Yo'LLaVA–Phi | 0.679 | 0.622 | — |
| DreamBooth (SD) | — | — | 0.066 | | DreamBooth (SD) | — | — | 0.012 |
| Yo'Chameleon | 0.489 | 0.495 | 0.279 | | Yo'Chameleon | 0.670 | 0.623 | 0.272 |
| UniCTokens (ours) | 0.527 | 0.601 | 0.371 | | UniCTokens (ours) | 0.700 | 0.683 | 0.352 |

Table 7: LLM and human scores on our bench. PARG = Personalized Attribute-Reasoning Generation.

## H   Limitation and Discussion

While our method demonstrates notable strengths, it is not without limitations. The first limitation is that, similar to other personalized models, our approach is constrained by the inherent limitations of the base model. Show-o struggles to generate outputs in varying styles, as illustrated in Fig. 8. Consequently, our method inherits the limitations of its underlying model. The second limitation pertains to the model's inability to effectively handle domain shift inputs, particularly in specialized fields such as medicine. However, this challenge presents an opportunity for improvement, as the development of unified models could enhance generalization and address this issue in future work. Finally, although we have attained state-of-the-art results (e.g., achieving a facial similarity of 0.334) in personalized individual generation compared with other unified models, there remains a gap when it comes to personalizing human faces. For reference, the recommended threshold for facial recognition similarity is around 0.4~0.5, representing a significant avenue for further research.

## I   Broader Impacts

The development of UniCTokens and the accompanying UnifyBench benchmark holds significant potential for advancing the field of personalized AI. By effectively integrating user-provided concepts into a unified vision language model, our approach not only enhances the performance of personalized understanding and generation tasks but also opens up new avenues for applications across various domains, such as education, healthcare, and creative industries. The ability to generate contextually relevant images based on minimal prompts can greatly benefit creative professionals by streamlining content creation processes. Additionally, our research emphasizes the importance of mutual reinforcement between understanding and generation, which could lead to more intuitive human-AI interactions. As we release our code and dataset, we aim to foster further research in this area, encouraging the development of more sophisticated models that can better understand and respond to user needs while ensuring ethical considerations in AI deployment.

## 🐏 Animal

| **<bo>** | **<mam>** | **<maeve>** |
|---|---|---|
|  |  |  |
| • <bo> is a Shiba Inu dog with a round face, fluffy cream and reddish-brown fur, and a curly tail.
• <bo> has a red hat.
• <bo>'s resting spot is on a soft white carpet. | • <mam> is a British Shorthair cat with silver tabby fur, large round amber eyes, and a stocky build.
• <mam> has a red hat.
• <mam>'s home is a cozy wooden cabin. | • <maeve> is a small white dog with fluffy fur, a bushy tail, and distinctive dark markings around the eyes and ears.
• <maeve> wears a pink bib when going out.
• <maeve> rests on a sofa indoors. |

## 👫 Person

| **<N>** | **** | **<W>** |
|---|---|---|
|  |  |  |
| • <N> is a young woman with long black hair, porcelain skin, and delicate facial features often highlighted by stylish makeup.
• <N> likes to wear white athletic short-sleeved shirts. | •  is a man with a close-cropped beard, medium-dark skin, and a clean fade haircut.
•  owns a dog as his pet.
•  has a pair of diamond stud earrings. | • <W> is a well-groomed man with neatly styled black hair, fair skin, and a defined jawline.
• <W>'s home is by the sea.
• <W> wears subtle lapel pins as accessories on his jackets. |

| **<WN>** | **<WL>** | **<C>** |
|---|---|---|
|  |  |  |
| • <WN> is a cheerful young man with short wavy brown hair and fair skin.
• <WN> enjoys eating street food while casually dressed.
• <WN> has a blue T-shirt. | • <WL> is a charismatic man with a well-groomed beard, short black hair, and a radiant smile.
• <WL> frequently appears at film premieres dressed in dark-colored suits.
• <WL> has a yellow hat. | • <C> is a professional tennis player known for her athletic build and signature braided hairstyle.
• <C> often wears a purple tank top on the court.
• <C> owns a pearl necklace. |

## ☕ Object

| **<pg>** | **** | **<nha>** |
|---|---|---|
|  |  |  |
| • <pg> is a ceramic mug designed like a pig's face, featuring round ears, pink blush cheeks, and a lid topped with a sculpted peach.
• <pg> belongs to a young girl. | •  is a circular red, white, and blue shield with a metallic finish and a star at the center.
•  appears rusty when exposed to rain. | • <nha> is a Gothic Revival cathedral with twin bell towers, pointed arches, and a large rose window on the facade.
• <nha> is decorated with flowers and banners during festivals. |

Figure 11: **UnifyBench Dataset.** Example images for partial concept in our constructed dataset.

