# OpenReview forum: "UniCTokens: Boosting Personalized Understanding and Generation via Unified Concept Tokens"
_NeurIPS.cc/2025/Conference — NeurIPS 2025 poster_

### Official Review · Reviewer_MqL1 · 2025-07-02

**Clarity:** 2
**Significance:** 3
**Originality:** 2
**Rating:** 4
**Confidence:** 4

**Summary:**

This paper introduces UniCTokens, a novel personalization framework that enhances both personalized generation and understanding in a unified VLM by fine-tuning unified concept tokens. Unlike prior methods that treat these two tasks independently, UniCTokens enables mutual enhancement by sharing learned representations across tasks. The authors also propose UnifyBench, a new benchmark designed to evaluate concept understanding, concept generation, and knowledge-driven generation within a unified framework. Experimental results on UnifyBench demonstrate the effectiveness and efficiency of the proposed approach.

**Questions:**

While UniCTokens enables efficient personalized understanding and generation with only a few training samples per concept, the framework seems to require training for each new concept. How scalable is this approach in scenarios where a large number of personalized concepts need to be integrated? Additionally, what are the computational and time costs associated with adding new concepts?

**Ethical Concerns:**

["NO or VERY MINOR ethics concerns only"]

**Final Justification:**

The authors addressed most of my concerns. I would like to remain my score with borderline accept.

**Limitations:**

Please refer to "weakness".

**Quality:**

3

**Strengths And Weaknesses:**

Strengths

-	This paper proposes a new framework for personalizing unified VLMs by fine-tuning unified concept tokens for both understanding and generation.
-	The proposed UnifyBench benchmark seems to contribute to the evaluation of concept understanding, concept generation, and knowledge-driven generation of a personalized unified model.
-	UniCTokens achieves state-of-the-art performance while maintaining a smaller model size and requiring fewer training samples.

Weaknesses

-	The term “Personalized Knowledge-Driven Generation” might be misleading. As presented, it largely refers to image generation guided by personalized textual attributes (e.g., "⟨x⟩ wearing its hat"), rather than a more complex or knowledge-integrative reasoning process. Clarifying this terminology would improve reader comprehension.
-	The presentation of results lacks clarity in Table 1. The content (L208-214, L216-219) suggests that it includes results for "Personalized Concept Generation" and "Personalized Knowledge-Driven Generation", but only "Personalized Generation" appear to be explicitly reported in this table.
-	The paper does not address how the model handles prompts that include multiple concepts (e.g., “⟨x⟩ and ⟨y⟩ sitting together”), which is a common scenario in real-world personalization tasks. Including an analysis or discussion of multi-concept generation would strengthen the work.

---

> ### Author Rebuttal · Authors · 2025-07-31
>
> ## Response to Official Review by Reviewer MqL1
> **Rebuttal:**
>
> Thank you very much for taking the time to review our paper. After carefully considering the feedback, we offer the following detailed responses organized point-by-point:
>
> > W1: The term “Personalized Knowledge-Driven Generation” might be misleading. Clarifying this terminology would improve reader comprehension.
>
> Thank you for dedicating your time to reviewing our paper and pointing out. We will clarify the terminology after introducing the newly defined task to avoid potential misleading. We propose renaming "Personalized Knowledge-Driven Generation" to "Personalized Attribute-Driven Generation" to emphasize the core of the task, which may mitigate potential misunderstandings. If you have any better suggestions for naming, we would be glad to discuss them with you.
>
> > W2: Only "Personalized Generation" appear to be explicitly reported in this table.
>
> Thank you for highlighting this issue. In fact, in Tables 1 and 3, we placed "Personalized Knowledge-Driven Generation" under the "T2I" column of "Personalized Generation", which may have caused some misleading. Overall, your suggestion is highly valuable and practical. We will revise the table structure based on the corrected descriptions in W1 to avoid further misunderstandings.
>
> > W3: The paper does not address how the model handles prompts that include multiple concepts.
>
> First, the current unified model has limitations when it comes to generating multiple concepts, which is orthogonal to our proposed method. This limitation has been explicitly discussed in the appendix of Yo'Chameleon[1]. Due to the NIPS Rebuttal policy, which prohibits the inclusion of images or links in the rebuttal, we designed a simple experiment to demonstrate that Show-o shares the same shortcoming in generating multiple concepts. We utilized GPT-4 to generate 100 prompts containing multiple concepts, consistent with the types in the Unify Bench, and allowed Show-o to generate outputs directly. To evaluate the ability to generate multiple concepts simultaneously, the metric is straightforward: a score of 1 is given if multiple concepts are generated simultaneously; otherwise, a score of 0 is assigned. To assess the quality of the generated images, we also had GPT score the outputs. We have listed the performance below:
>
> | Model | GPT |
> |:---|:---:|
> |Show-o| 0.37|
> |Show-o2-7B| 0.51 |
> |Chameleon | 0.21 |
>
> It can be observed from the table that both Show-o and Chameleon struggle to generate multiple concepts. Building on Show-o, UniCTokens also inherits these limitations. If better unified models become available in the future, our method has the potential to benefit from these advancements. Finally, we would like to clarify that the primary focus of our paper is on unified model personalization and facilitating mutual enhancement between understanding and generation tasks. We will consider addressing the issue of multiple concept generation in future work.
>
> > Q1: How scalable is this approach in scenarios where a large number of personalized concepts need to be integrated?
>
> For general fine-tuning based methods (e.g., Dreambooth, Yo'LLaVA), each concept has its own set of parameters. To evaluate the scalability of our method, we selected five concepts from the Unify Bench and employed a progressively continuous training approach. After learning one concept, we freeze the parameter and prepare new parameters for the next concept. After training on all the concepts, we conducted tests on each of them:
>
> |    Type           | Rec                        | VQA                        | QA                         | Pure Gen                | Pure Gen                | People Gen               | Knowldeg-driven        | Knowldeg-driven        |
> |:---------------|:----------------------------:|:----------------------------:|:----------------------------:|:-------------------------:|:-------------------------:|:-------------------------:|:-------------------------:|:-------------------------:|
> | Metric        | Weight                     | GPT                         | GPT                        | CLIP-I                  | CLIP-T                  | Face-Simi              | Score                  |  CLIP-I                  |
> | Yo'LLaVA-Phi   | 0.737                      | 0.377                      | 0.382                      | -                       | -                       | -                       | -                       | -                       |
> | Dreambooth    | -                          | -                          | -                          | 0.611                   | 0.259                   | 0.101                   | 0.050                    | 0.608                   |
> | Yo'Chameleon   | 0.742                      | 0.391                      | 0.433                      | 0.623                   | 0.187                   | 0.156                   | 0.199                    | 0.623                   |
> | UniCTokens    | 0.758                      | 0.422                      | 0.480                     | 0.691                   | 0.270                   | 0.221                   | 0.307                    | 0.694                   |
>
> Continuously integrating new concepts may lead to catastrophic forgetting. However, UniCTokens still outperforms Yo'LLaVA-Phi, Dreambooth and Yo'Chameleon in this setting, which may be attributed to the well-designed progressive training strategy.
>
> > Q2: What are the computational and time costs associated with adding new concepts?
>
> As you suggested, in accordance with the Q1 setting, we have presented the computational and time costs comparison results of adding new concepts. From this table, we can draw the following observations:
>
> | Type | FLOPs/Concept | Time/Concept |
> |:---|:---:|:---:|
> |Yo'LLaVA-13B|$1.5×10^{17}$|30min|
> |Yo'LLaVA-Phi| $1.6×10^{16}$ |10min|
> |Dreambooth(SD)| $2×10^{15}$ |7min|
> |Yo'Chameleon| $7×10^{17}$ |120min|
> |UniCTokens| $1.3×10^{17}$ |25min|
>
> It can be observed that Yo'Chameleon takes a long time because it trains each concept using approximately 1,100 images. In contrast, UniCTokens achieves stronger MMU and T2I capabilities while keeping the training computational and time costs under control.
>
> We sincerely hope the above explanations can alleviate your remaining concerns and any potential misunderstandings about our paper. And we will include all the above-mentioned tables in the revised version of our paper. We wish this work could bring new insights to the field of personalization for unified models and provide feasible solutions for cross-task transfer and mutual enhancement between tasks in more complex scenarios.
>
> [1] Yo'Chameleon: Personalized Vision and Language Generation

---

> > ### Comment · Reviewer_MqL1 · 2025-08-03
> >
> > Thank you for your detailed rebuttal. The responses addressed most of my concerns. Clarifying the task definition, improving the table presentation, and providing further discussion on multiple concept generation as well as the computational cost of adding new concepts will enhance the clarity and transparency of the paper. I encourage you to incorporate these points into the final version.

---

> > > ### Author Response · Authors · 2025-08-04
> > > **Official Comment by Authors**
> > >
> > > Thank you for your response. We will incorporate these points into the final version as your suggestion. If you have any remaining concerns, please let us know, and we’ll respond promptly. If you feel our work merits it, we’d be grateful if you could consider raising the rating.

---

### Official Review · Reviewer_Sy58 · 2025-07-02

**Clarity:** 2
**Significance:** 2
**Originality:** 3
**Rating:** 4
**Confidence:** 3

**Summary:**

The paper proposes UniCTokens, a framework integrating personalized information into a unified vision language model (VLM) for understanding and generation. The paper proposes a progressive training strategy with three stages: understanding warm-up, bootstrapping generation from understanding, and deepening understanding from generation to enhance mutual benefits between these tasks.  Moreover, it presents a benchmark UnifyBench to evaluate the unified VLM personalization.

**Questions:**

L44, mutual information, what does it mean?
The current evalution on several tasks, such VQA, how about other complex VL tasks, such as visual dialogue.

**Ethical Concerns:**

["NO or VERY MINOR ethics concerns only"]

**Final Justification:**

Some of my concerns have been addressed. I would like keep my score.

**Limitations:**

K-means is adopt for clustering. So some ad-hoc parameter is required.
The training of the model has three stages. So the reproducibility is a little in doubt.

**Paper Formatting Concerns:**

The text and figures in Tables is not in black, slightly not clear enough.

**Quality:**

3

**Strengths And Weaknesses:**

Pros
1. The unified concept tokens proposed in the paper enable learnable tokens to be simultaneously applied to both understanding and generation tasks, breaking the limitations of tasks' isolation.
2. The three-stage training method progressively enhances the model's understanding and generation capabilities while enabling mutual improvement between them.
3. The paper presents the first personalized dataset, which covers three categories: humans, pets, and objects.

Cons
1. Regarding Section 1.3.1 describing the training objectives of unified concept tokens M, provide more detailed differentiation between Objective 2 and Objective 3 to better highlight their differences and implementation challenges.
2. For image generation evaluation, introducing more challenging tests with noisy backgrounds (e.g., generating the target concept <bo> alongside similar category instances) would provide a more rigorous assessment of the model's concept retention and robustness in complex scenarios.

---

> ### Author Rebuttal · Authors · 2025-07-31
>
> ## Response to Official Review by Reviewer Sy58:
> **Rebuttal:**
>
> Thanks for your thoughtful response and the time you have invested in reviewing our paper. After carefully reviewing the comments, we present the following responses in a point-by-point format:
>
> > C1: Provide more detailed differentiation between Objective 2 and Objective 3.
>
> We sincerely apologize for any potential ambiguities. As shown in Figure 1, the commonality between Object 2 and Object 3 lies in their classification as image generation tasks that necessitate the production of high-quality personalized concepts. The distinction between them is that the prompt for Object 2 contains only the personalized concept, whereas the prompt for Object 3 additionally incorporates certain information that is only within the understanding data (e.g., "<bo> has a red hat" in understanding data, "a photo of <bo> wearing its hat" for knowledge-driven generation). Failure to leverage this information would preclude the generation of the hat in an appropriate color. Our objective is to utilize this task to measure the extent of information transfer across tasks. We will clarify this point in the revised version to avoid any misunderstandings.
>
> > C2: Introduce more challenging tests with noisy backgrounds.
>
> Thank you for your suggestion. This essentially represents a special case of multiple concept generation. In fact, personalization tasks often focus more on the consistency between the generated subject and the reference object. Meanwhile, there are currently no established benchmarks available for evaluation. We appreciate your advice and have manually constructed prompt templates such as "a photo of <sks> and a <cls>" for evaluating on Unified Bench. Here, the <cls> refers to the same category as <sks>. For example, we want to generate a Tom cat, the prompt will tend to be like "a photo of <Tom> and a cat". The results are as follows:
>
> | Method| Training images | Personalized Generation(CLIP-I) | Personalized Generation(CLIP-T) |
> |:-----|:---------------:|:-------------------------------:|:-------------------------------:|
> | Dreambooth(SD) | ~10 | 0.627 | 0.234 |
> | Textinv | ~10 | 0.600 | 0.222 |
> | Show-o+TP | - | 0.659 | 0.240 |
> | Yo'Chameleon | ~1000 | 0.691 | 0.207 |
> | UniCTokens | ~10 | 0.723 | 0.243 |
>
> It can be observed from the table that under such challenging and noisy background conditions, the performance of personalized generation is somewhat weakened. Nevertheless, our UniCTokens still achieved SOTA performance, demonstrating the effectiveness and robustness of our approach.
>
> > Q1: L44, mutual information, what does it mean?
>
> We apologize for any potential misunderstandings that the paper may have caused. Mutual information refers to information that can be mutually utilized across tasks. This is primarily manifested through the pre-training of understanding models, which leverage the implicit priors encoded in the model parameters to assist in generation, as well as utilizing intermediate results from the generation process to explicitly aid in fine-grained understanding. The specific results of this mutual enhancement are quantitatively presented in Table 3 of the main text. To express this more clearly, we will revise it to incorporate the term "mutual semantics". We welcome any suggestions you may have regarding the naming.
>
> > Q2: How about other complex VL tasks, such as visual dialogue.
>
> Thank you for your insightful question. To the best of our knowledge, none of the existing works[1,2,3] have mentioned this type of evaluation, and there are currently no established benchmarks available for this evaluation. However, to address your concern, we created a small-scale testing set that includes two multi-turn dialogues for each concept generated by GPT, with each dialogue consisting of 2 to 3 turns. The rationale behind this construction is to use GPT to generate multiple questions about the concept in the images, selecting question pairs that can form a dialogue and linking them to create a coherent conversation, thereby addressing the evaluation gap.
>
> | Method | Backbone | Visual Dialogue(BLEU) | Visual Dialogue(GPT) |
> |:------:| :---:|:---:|:---:|
> | Show-o+TP | Show-o | 0.391 | 0.388 |
> | UniCTokens | Show-o | 0.459 | 0.450 |
>
> It can be observed that the performance of the unified model backbone declines on complex VL tasks. Additionally, performance of UniCTokens also drops when building upon Show-o. Complex VL tasks present both a significant challenge and a promising research direction, and we plan to explore solutions in future work.
>
> > L1: Some ad-hoc parameter is required; So the reproducibility is a little in doubt.
>
> Thank you for pointing this out. In fact, we place a high priority on reproducibility, and have presented several hyperparameters and training details in Table 4 of the appendix. We will include more hyper parameter setup to further ensure the reproducibility of our results. Regarding the aforementioned ad-hoc parameter in K-means for identifying the most valuable visual tokens, our experimental setup involves setting K to 2, using cosine similarity as the distance metric, where we select the cluster with the highest number of retained patches.
>
> [1] MyVLM: Personalizing VLMs for User-Specific Queries
>
> [2] Yo'LLaVA: Your Personalized Language and Vision Assistant
>
> [3] RAP: Retrieval-Augmented Personalization for Multimodal Large Language Models

---

> > ### Comment · Reviewer_Sy58 · 2025-08-05
> >
> > Thank you for your comprehensive responses addressing my key concerns. Adding even clearer descriptions and expanded ablation studies promises to significantly improve the paper's clarity and solidity. I strongly encourage you to implement these enhancements for the final version.

---

### Official Review · Reviewer_mWTp · 2025-07-03

**Clarity:** 3
**Significance:** 3
**Originality:** 3
**Rating:** 5
**Confidence:** 2

**Summary:**

This paper introduces UniCTokens, a novel framework for personalizing a unified VLM to handle both understanding and generation tasks for user-provided concepts. The key idea is to move away from separate concept tokens for each task and instead train a single, unified set of concept tokens. A core contribution is a three-stage progressive training strategy. To evaluate their method, the authors also introduce UnifyBench, a new benchmark designed to assess personalized understanding, generation, and a new task they define as "knowledge-driven generation," where the model must generate images combining visual concepts with attributes described only in text.

**Questions:**

I'm mostly curious about the complexity and scalability. Is the three-stage process truly necessary, or could a simpler joint-training scheme work? Meanwhile, beside 1.3B model, have you tried on 7B Show-o?

**Ethical Concerns:**

["NO or VERY MINOR ethics concerns only"]

**Final Justification:**

The authors addressed my concerns during the rebuttal. I believe that incorporating the experiments presented in the rebuttal into the final version will strengthen the paper.

**Limitations:**

Yes.

**Quality:**

3

**Strengths And Weaknesses:**

This paper presents a novel approach to personalization in unified VLMs by using a single set of concept tokens for both understanding and generation. This contrasts with prior work like Yo’Chameleon and enables more effective cross-task transfer, particularly for the “knowledge-driven generation” task. The three-stage training strategy is technically sound and the feedback loop introduced in Stage 3 is especially insightful.

Experiments are thorough, with strong baselines and a new benchmark (UnifyBench) supporting the claims. Ablations clearly show the benefit of each component.

However, the framework adds complexity by introducing multiple token types and training phases, and it's unclear if a simpler setup could achieve similar results. The model scale is relatively small (1.3B), so it's uncertain how well the approach generalizes. Also, evaluation relies heavily on GPT-4o, which may introduce bias.

---

> ### Author Rebuttal · Authors · 2025-07-31
>
> ## Response to Official Review by Reviewer mWTp:
> **Rebuttal:**
>
> We sincerely appreciate your recognition of our paper and your valuable comments. We are encouraged by the recognition that our method is novel and insightful. After thoroughly reading the comments, we provide the following point-by-point responses：
>
> > W1 & Q1: It's unclear if a simpler setup could achieve similar results; Is the three-stage process truly necessary, or could a simpler joint-training scheme work?
>
> To further elucidate the necessity of the three-stage strategy, we conducted additional exploratory experiments. Our experimental setup is as follows: To ensure a fair comparison, we maintained the same total number of learnable parameters and training data as in the UniCTokens, but directly executed single-stage joint training. For your convenience in comparing, we present the results of the single-stage approach alongside those of UniCTokens as follows:
>
> |   Task Type     |     Rec   |   VQA   |   QA    | Pure Gen | Pure Gen | People Gen | Knowledge-driven | Knowledge-driven |
> |:---------------|:-------:|:-------:|:-------:|:--------:|:--------:|:----------:|:----------------:|:----------------:|
> |     Metric      | Weight  |  GPT    |  GPT    |  CLIP-I  |  CLIP-T  | Face-Simi  |      Score       |      CLIP-I      |
> | Joint Training  | 0.709   | 0.489   | 0.565   |  0.681   |  0.258   |   0.288    |      0.269       |      0.678       |
> | 3-Stage(Ours)   | 0.790   | 0.523   | 0.603   |  0.750   |  0.282   |   0.334    |      0.359       |      0.749       |
>
> As can be observed, our UniCTokens outperforms joint training across personalized understanding and generation tasks, especially on knowledge-driven generation. We believe that it is our three-stage training that facilitates better cross-task information transfer, and this paradigm is superior to direct joint training.
>
> > W2 & Q2: It's uncertain how well the approach generalizes; Meanwhile, beside 1.3B model, have you tried on 7B Show-o?
>
> Thank you for your insightful question and suggestion. First, we would like to clarify that when we completed this work, the Show-o2-7B model had not yet been released (2025.06.19). Second, to validate the generalizability of our method, we supplement the results of UniCTokens on Show-o2-7B with the learnable parameters and training data remained consistent with Show-o in the main text:
>
> |     Task Type     |   Rec  |       VQA      |       QA      | Pure Gen | Pure Gen | People Gen | Knowledge-driven | Knowledge-driven |
> |:-----------------|:-------:|:--------------:|:-------------:|:------------------:|:---------------:|:----------------:|:----------------:|:----------------:|
> |      Metric       | Weight  |      GPT       |     GPT       |   CLIP-I  | CLIP-T |   Face-Simi     |      Score       |      CLIP-I      |
> | UniCTokens(Show-o2-7B)    | 0.857   |     0.595      |    0.679      |   0.799   | 0.297  |     0.381       |      0.372       |      0.801       |
> | UniCTokens(Show-o)  | 0.790   |     0.523      |    0.603      |   0.750   | 0.282  |     0.334       |      0.359       |      0.749       |
>
> When the model is scaled, the performance on the task is observed to improve correspondingly. Our approach is designed to be compatible with the base model; it is evident that the improvement of the base model's capability leads to a significant enhancement in the results produced by our method, thereby demonstrating the generalizability of our approach across models of diverse scales.
>
> > W3: Also, evaluation relies heavily on GPT-4o, which may introduce bias.
>
> Thank you for pointing this out. We would like to clarify two key points regarding our evaluation methodology:
>
> 1. **More Metrics Utilized:** We employed not only GPT-4o for evaluation in our main text but also utilized several classical metrics for understanding and generation (e.g., BLEU, CLIP-I). GPT scoring also increasingly becomes a widely accepted evaluation method in both general[1] and personalization[2] scenarios. However, to further mitigate potential biases in evaluation, we employed Gemini-2.5-Pro as a judge for the extra evaluation on the Unify Bench.
> 2. **Human Study:** To ensure that the evaluation results of large models align with human preferences, we also sampled 300 results from the test set (each concept has 5 samples for each task) and recruited professionals from technology companies and college students to conduct a detailed human study. The scoring range for human graders is consistent with that of large language models. We compare against several baselines:
>
> |    Methods      | Training images |   VQA (Gemini)   |   VQA (Human)   |   QA (Gemini)   |   QA (Human)   | Knowledge-driven (Gemini) | Knowledge-driven (Human) |
> |:---------------|:--------------:|:----------------:|:---------------:|:---------------:|:--------------:|:-------------------------------:|:------------------------:|
> | Yo'LLaVA-Phi     |     ~100       |      0.492       |      0.679      |      0.498      |     0.622      |                -                |            -             |
> | Dreambooth(SD)  |     ~10        |        -         |        -        |        -        |       -        |              0.066              |         0.012            |
> | Yo'Chameleon    |     ~1000      |      0.489       |      0.670      |      0.495      |     0.623      |              0.279              |         0.272
> | UniCTokens      |     ~10        |      0.527       |      0.700      |      0.601      |     0.683      |              0.371              |         0.352            |
>
> In comparison with Table 1 in the main text, the trends of Gemini and GPT scores are consistent and are generally aligned with the results from the human study. We will add extra metrics in the revised version to ensure unbiased results.
>
> Considering the limited time during the rebuttal period, we will include more comparisons on understanding-only and generation-only benchmarks of the above mentioned experiments in the revised version.
>
> [1] Judging LLM-as-a-Judge with MT-Bench and Chatbot Arena
>
> [2] MC-LLaVA: Multi-Concept Personalized Vision-Language Model

---

> ### Comment · Reviewer_mWTp · 2025-08-01
>
> Thanks for the rebuttal. The authors addressed my concerns during the rebuttal. I believe that incorporating the experiments presented in the rebuttal into the final version will strengthen the paper. I'm raising my rating from 4 to 5.

---

> > ### Author Response · Authors · 2025-08-02
> > **Official Comment by Authors**
> >
> > Dear Reviewer mWTp,
> >
> > We sincerely thank you for recognizing the merits of our paper. I am glad that our response resolves your problems! Your support means a lot to us!
> >
> > Best wishes,
> >
> > Authors

---

### Decision · Program_Chairs · 2025-09-17

**Decision:**

Accept (poster)

**Comment:**

All three reviewers recommend acceptance of this submission, highlighting the strengths of its framework and method. While no major issues remain, their acceptance is conditioned on the authors incorporating the changes promised in the rebuttal, specifically including, adding the new experiments, clarifying the task definition, improving table presentation, expanding the discussion on multiple concepts, and providing clearer descriptions along with additional ablation studies.